# Global Surveillance and Biological Characterization of the SARS-CoV-2 NB.1.8.1 Variant: An Emerging VUM Lineage Under Scrutiny

**DOI:** 10.3390/v17111457

**Published:** 2025-10-31

**Authors:** Gaojie Cao, Chenhui Xu, Linxi Wang, Keikei Chai, Beibei Wu

**Affiliations:** 1School of Laboratory Medicine and Life Science, Wenzhou Medical University, Wenzhou 325035, China; caogaojie0111@163.com (G.C.); xuchenhui0521@163.com (C.X.); 2School of Public Health, Hangzhou Medical College, Hangzhou 310051, China; wanglx9923sjk@163.com; 3School of Public Health and Management, Wenzhou Medical University, Wenzhou 325035, China; 13386604861@163.com; 4Zhejiang Provincial Center for Disease Control and Prevention, Hangzhou 310051, China

**Keywords:** SARS-CoV-2, NB.1.8.1, variant monitoring, immune evasion

## Abstract

The continuous evolution of SARS-CoV-2 and its variants poses persistent challenges to global public health. As a sublineage of the XDV.1 variant, NB.1.8.1 has rapidly emerged as a dominant strain worldwide, triggering a new wave of infections. Representing a product of viral adaptation, this variant has acquired several critical amino acid mutations—including A435S and T478I—which enhance its transmissibility and immune evasion capabilities compared to the ancestral XDV.1 lineage. This review systematically summarizes the genomic characteristics, epidemiological features, and immune escape potential of NB.1.8.1. It emphasizes that sustained genomic surveillance and serological assessments are crucial for informing public health response strategies, guiding vaccine development, and optimizing containment measures.

## 1. Introduction

Since its emergence in late 2019, *SARS-CoV-2* (Severe Acute Respiratory Syndrome Coronavirus-2), the causative agent of the COVID-19 pandemic, has demonstrated a remarkably dynamic and rapid evolutionary pathway, continuously challenging global public health systems and prompting an unprecedented scientific response [1,2,3]. As a member of the Betacoronavirus genus, within the Coronaviridae family [4,5,6], this enveloped virus contains a positive-sense single-stranded RNA genome approximately 30 kilobases in length, encoding both structural and non-structural proteins critical for its replication cycle and pathogenesis. In particular, the spike protein is closely associated with viral entry into host cells, serves as a critical site for mutations in subsequent variants, and represents an important target for vaccine development [7,8,9,10] (see Figure 1). The replication of the virus is mediated by an RNA-dependent RNA polymerase (RdRp) complex that exhibits intrinsically low fidelity due to the absence of proofreading functionality [11,12], a common trait among RNA viruses [13]. *SARS-CoV-2* possesses a unique replication proofreading function, mediated by the exoribonuclease activity of its NSP14 protein, to correct mismatched nucleotides. This fundamental molecular characteristic facilitates the accumulation of mutations—including substitutions, insertions, and deletions—during viral replication and transmission within human populations. This process fosters significant genetic diversity and provides the raw material for natural selection, continually giving rise to numerous variants with altered phenotypic and antigenic profiles, which are characterized by differences in transmissibility, pathogenicity, and ability to evade host immunity.

The evolutionary process of SARS-CoV-2 is principally governed by diverse and interacting natural selection pressures exerted by a range of factors at both the individual host and population levels. Key among these are host immune responses, including neutralizing antibodies targeting primarily the spike (S) protein—especially the receptor-binding domain (RBD)—and T-cell-mediated immunity directed against various viral epitopes. Furthermore, pre-existing population immunity, resulting from prior infection(s) with different variants or from vaccination using various platforms (e.g., mRNA, viral vector, inactivated virus), creates a complex immunological landscape [14,15,16,17]. This often leads to immune imprinting, where prior antigenic exposure shapes and potentially restricts subsequent immune responses to new variants. Additional selective forces include transmission dynamics within populations with varying levels of immunity and contact patterns, environmental stability of the virus, and potential zoonotic adaptations or spillback events involving animal reservoirs. In the context of extensive global vaccination campaigns and recurrent waves of infection, SARS-CoV-2 evolution has been strikingly marked by significant convergent evolution [18,19,20,21]. This phenomenon is observed when distantly related lineages independently acquire identical mutations to overcome similar selective hurdles, particularly those posed by population immunity. This is most evident in key antigenic regions of the viral spike (S) protein, which mediates host cell entry via the angiotensin-converting enzyme 2 (ACE2) receptor. Specific domains within the S protein, such as the receptor-binding domain (RBD) and the N-terminal domain (NTD), have emerged as recurrent mutational hotspots. Certain critical residues, including K417, L452, E484, F486, Q493, and N501, have undergone repeated, independent mutations across numerous emerging lineages [22,23,24]. These specific amino acid changes are frequently associated with improved viral fitness traits [25], such as enhanced binding affinity to the human ACE2 receptor, increased transmissibility through various mechanisms, and a significantly increased capacity to evade neutralizing antibodies, thereby undermining humoral immunity.

In addition to point mutations, recombination has emerged as a crucial and increasingly recognized mechanism driving the rapid evolution of SARS-CoV-2, especially as the global population developed high levels of immunity from mixed exposures. Genetic recombination between distinct co-circulating variants infecting the same individual can enable the virus to rapidly combine advantageous mutations from different parents into a single genome, potentially creating viruses with novel combinations of fitness-enhancing mutations [26,27,28]. For example, the XBB variant, which demonstrated substantial immune escape, originated from recombination between BA.2.10.1 and BA.2.75 sublineages. More recently, recombinant variants such as XDV (a recombinant of XDE and JN.1) and NB.1.8.1 (specifically XDV.1.5.1.1.8.1) have been identified and gained prevalence, illustrating the critical role of recombination in generating viruses with potentially enhanced adaptability, complicating the prediction of future evolutionary trajectories.

To systematically monitor, assess, and communicate the public health risk posed by these continuously emerging variants, the World Health Organization (WHO) has established and continually refined a structured framework classifying viral variants into three main categories: Variants of Concern (VOCs), Variants of Interest (VOIs), and Variants Under Monitoring (VUMs) [29,30]. This risk assessment integrates data on transmissibility, disease severity, immune escape, and the potential impact on available countermeasures like diagnostics, therapeutics, and vaccines. Since the original Wuhan-Hu-1 strain, several VOCs have sequentially emerged, each markedly altering the trajectory of the pandemic and often outcompeting previous circulating viruses due to selective advantages. These include Alpha (B.1.1.7), associated with increased transmissibility; Beta (B.1.351), which showed significant immune evasion properties; Gamma (P.1); and Delta (B.1.617.2), which was characterized by very high viral loads and increased severity [31,32,33]. The Omicron variant (B.1.1.529), characterized by an unprecedented number of mutations, particularly in the spike protein (over 30 changes), eventually displaced all previous VOCs due to its superior transmissibility and profound immune evasion properties, leading to massive infection waves globally. Omicron itself has since diversified into a vast constellation of multiple sublineages, such as BA.2, BA.4, BA.5, BQ.1, XBB, and JN.1 (a descendant of BA.2.86), which dominated global incidence for many months. Current surveillance data from platforms like GISAID indicate that lineages descending from NB.1.8.1 are becoming predominant across various regions [34,35], highlighting the virus’s relentless evolution.

Although current epidemiological evidence does not clearly associate NB.1.8.1 and its sublineages with substantially increased intrinsic disease severity or a substantial surge in global hospitalization and mortality rates, this variant exhibits a notable growth advantage and increased effective reproduction number (Re) in multiple regions compared to contemporaneous lineages [36]. This suggests ongoing viral adaptation and optimization within the human population. This adaptation may be driven not only by spike mutations but also by mutations in non-structural proteins (e.g., those involved in the replication-transcription complex) or accessory proteins that could modulate viral replication efficiency, pathogenesis, or antagonism of the host innate immune response. The evolutionary trajectory of SARS-CoV-2 remains inherently unpredictable. Future variants may exhibit further enhancements in transmissibility, immune evasion (against both natural and vaccine-induced immunity), or altered tissue tropism, which could potentially change clinical manifestations. Given this constant and evolving threat, maintaining a robust, sensitive, and coordinated global genomic surveillance system is absolutely imperative. This requires continued sequencing efforts, rapid data sharing through public repositories like GISAID, and advanced bioinformatic analysis. Integrating comprehensive genomic data with detailed molecular epidemiological investigations, thorough phenotypic characterization (e.g., through pseudovirus neutralization assays, plaque reduction neutralization tests, live virus studies, binding affinity studies using Surface Plasmon Resonance, and animal model studies), and systematic serological assessments is essential for the early detection and risk assessment of high-risk variants, the timely evaluation of vaccine effectiveness, and the evidence-based development and deployment of targeted public health interventions, including updated vaccine formulations and therapeutic antibodies.

The purpose of this review is to synthesize the currently available evidence on the global transmission dynamics, genomic characteristics, and phenotypic properties of the SARS-CoV-2 variant NB.1.8.1, in order to systematically evaluate its potential public health risk. By integrating data from genomic surveillance databases, in vitro phenotypic studies, and emerging epidemiological reports, this work aims to clarify the variant’s origin, mutational profile, fitness advantages, and immune evasion capabilities. Special emphasis is placed on assessing its transmissibility relative to other circulating variants, its potential impact on vaccine effectiveness, and its role in possible resurgence events. Ultimately, this comprehensive analysis seeks to identify key knowledge gaps, inform risk assessment efforts, and provide a scientific basis for public health guidance and intervention strategies.

## 2. The Backdrop of JN.1 Global Dominance and the Emergence of the Novel NB.1.8.1 Lineage

The SARS-CoV-2 virus is the etiological agent responsible for the COVID-19 pandemic. Its pronounced mutability and evolutionary capacity have led to the emergence of numerous variants, which have significantly influenced the trajectory of the pandemic. Among these, the Omicron variant (B.1.1.529) and its sublineages have dominated globally since late 2021. The JN.1 variant, a descendant of the BA.2.86 lineage (which itself belongs to the Omicron family), emerged as a significant Variant of Concern (VOC) by the end of 2023 and rapidly became the predominant strain worldwide in early 2024 [37,38,39]. Its emergence underscores the adaptive evolution of the virus, characterized by enhanced transmissibility, immune evasion capabilities, and a constellation of unique mutations that facilitated its global dissemination.JN.1 (designated as BA.2.86.1.1) originated from the BA.2.86 lineage [40], which in turn evolved from the earlier Omicron BA.2 variant. BA.2.86 attracted attention due to its extensive mutational profile, featuring over 30 amino acid changes in the spike protein compared to BA.2, with notable concentrations in the receptor-binding domain (RBD) and the N-terminal domain (NTD). These mutations represent a “step-change” in viral evolution, analogous to the initial emergence of Omicron, leading to significant antigenic divergence from preceding variants [41,42,43]. JN.1 was first detected in Luxembourg in August 2023 and quickly spread to multiple countries. By November 2023, its global prevalence increased markedly, rising from approximately 4% of cases to around 30% by December 2023. This rapid expansion prompted the World Health Organization (WHO) to reclassify BA.2.86 from a Variant Under Monitoring (VUM) to a Variant of Interest (VOI) in November 2023 [44,45], with particular attention directed at JN.1 due to its transmission advantage. The defining feature of JN.1 is the L455S mutation within the RBD of the spike protein, which distinguishes it from its parental BA.2.86 lineage. This single substitution significantly enhances the variant’s immune evasion and infectivity. Studies indicate that L455S reduces the binding affinity of neutralizing antibodies, enabling JN.1 to circumvent immunity conferred by prior infection and vaccination [41]. Furthermore, structural analyses reveal that L455S alters the hydrogen-bonding network with the human ACE2 receptor, thereby improving the efficiency of viral cell entry. This effect is particularly evident in human nasal epithelial cells (hNECs) [46], where JN.1 demonstrates superior infectivity compared to other contemporaneous variants such as BA.2.86 and XBB. In addition to L455S, JN.1 has inherited a suite of mutations from BA.2.86, including alterations in non-spike proteins, collectively contributing to its overall fitness. For instance, the reversion mutation R493Q in the RBD partially restores receptor-binding affinity [47], while other mutations such as K356T introduce novel N-glycosylation sites, further shielding the virus from antibody recognition. The global dissemination of JN.1 was facilitated by its ability to outcompete other circulating variants, including XBB and EG.5. Its growth advantage was observed across multiple regions, including Europe, the Americas, and Asia. By early 2024, JN.1 had become the dominant lineage worldwide, driving increases in case numbers despite pre-existing population immunity. The success of JN.1 also led to the emergence of sublineages such as KP.2, KP.3, JN.1.7, JN.1.18, and KP.3.1.1, many of which retain the critical L455S mutation while accumulating additional changes [48,49,50,51]. These sublineages exhibit even greater immune evasion properties. For example, KP.2 and KP.3, which also carry mutations such as F456L and Q493E, demonstrate increased resistance to neutralizing antibodies, including those induced by XBB.1.5-based vaccines. Among these, the KP.3.1.1 sublineage has drawn attention due to its potent immune evasion, explaining its rapid growth advantage in diverse populations. JN.1 and its sublineages pose considerable challenges to existing immunity. Serological studies indicate that JN.1 can evade antibodies generated by prior Omicron infections (e.g., XBB) and vaccination. For instance, convalescent sera from XBB-infected individuals exhibit reduced neutralization titers against JN.1, with a 2.6- to 3.8-fold decrease compared to other variants [52]. This immune evasion is attributed to the cumulative effect of RBD mutations, which alter key antigenic sites targeted by antibodies. The substantial antigenic shift from XBB to JN.1 has led to recommendations for updating vaccine formulations to include JN.1 or its sublineages. Evidence suggests that JN.1-specific infection can elicit a broader immune response, highlighting the importance of variant-specific vaccines in maintaining efficacy [53]. Despite its increased transmissibility, JN.1 is not associated with greater disease severity compared to earlier variants. Epidemiological data from multiple countries, including the United States and China, indicate that JN.1 infections are predominantly mild or asymptomatic [54,55]. This observation is supported by animal studies showing lower viral loads in the lungs and reduced tissue damage compared to XBB sublineages [56]. Nevertheless, the ability of JN.1 to infect individuals with prior immunity has resulted in increased case numbers, imposing additional burdens on healthcare systems. The rise in JN.1 as a major VOC and the subsequent emergence of its sublineages illustrate the ongoing evolutionary arms race between SARS-CoV-2 and human immunity. Although no increase in severity has been observed, the variant’s public health impact underscores the necessity of vigilant genomic surveillance, updated vaccines, and adaptive public health strategies. As the virus continues to evolve, understanding the mechanisms driving variants like JN.1 will be crucial for managing future waves of infection.

The initial identification of the SARS-CoV-2 variant NB.1.8.1 and its subsequent designation as a Variant Under Monitoring (VUM) underscore critical gaps in our current understanding of its origin, biological behavior, and potential public health impact. First detected in late 2024 through genomic surveillance systems, NB.1.8.1, a descendant of the recombinant XDV lineage, attracted attention due to its rapid expansion in certain regions despite low global prevalence. For instance, areas like Southeast Asia and California, USA. Its genetic profile includes multiple spike mutations, such as A435S and K478I, which are suggestive of possible enhanced infectivity and immune evasion. However, the precise evolutionary pathway leading to its emergence—whether through gradual accumulation of mutations or additional recombination events remains unclear. Furthermore, the functional implications of its mutational repertoire are not yet fully characterized. While pseudovirus assays indicate increased infectivity, real-world data on transmissibility, disease severity, and immune escape are limited. In particular, the effectiveness of existing vaccines and therapeutic monoclonal antibodies against NB.1.8.1 requires thorough investigation. Another significant knowledge gap involves the variant’s sensitivity to population immunity shaped by prior exposure to JN.1 and other recent variants. The absence of robust serological and epidemiological studies impedes risk assessment and evidence-based public health planning. Thus, while its current designation as a VUM reflects precautionary principles, filling these knowledge gaps through coordinated laboratory and clinical research is essential to determine whether NB.1.8.1 poses a substantial threat to global health.

## 3. Methods

To comprehensively identify all studies and genomic reports related to the NB.1.8.1 variant, we conducted a systematic literature search. Electronic databases including PubMed, Web of Science, and Scopus were searched from December 2024 to September 2025, with no language restrictions. We specifically queried the genomic sequence data and related metadata in the GISAID database, with the query parameters configured as follows: the lineage was set to NB.1.8.1 or XDV.1.5.1.1.8.1, and the corresponding time frame was defined as detailed below. The search strategy incorporated a combination of controlled vocabulary and keywords, including: “NB.1.8.1”, “SARS-CoV-2 variant”, “recombinant variant”, “spike mutation”, “immune evasion”, “Variant Under Monitoring”, and “VUM”. Boolean operators (AND/OR) were used to refine the search, such as: (“NB.1.8.1” OR “XDV.1.5.1.1.8.1”) AND (“spike mutation” OR “immune evasion” OR “transmissibility”). In addition, official public health reports from the World Health Organization (WHO), the European Centre for Disease Prevention and Control (ECDC), and the U.S. Centers for Disease Control and Prevention (CDC) were manually reviewed to supplement background and epidemiological information.

## 4. The Origin and Evolution of the NB.1.8.1 Lineage

The phylogenetic designation of the SARS-CoV-2 variant NB.1.8.1 reflects a complex evolutionary history shaped by multiple recombination events [57]. According to the World Health Organization (WHO) and multiple genomic studies, NB.1.8.1 is a direct descendant of the XDV.1.5.1 lineage, with the full Pango lineage designation XDV.1.5.1.1.8.1. This classification places it within a sophisticated phylogenetic network dominated by recombination dynamics. The XDV lineage itself is not a product of simple clonal evolution but represents a recombinant chimera derived from multiple ancestral sources. Specifically, XDV originated from recombination between the XDE and JN.1 lineages. Notably, XDE is itself a recombinant, arising from two XBB sublineages—GW.5.1 and FL.13.4. (see Figure 2) Consequently, the genome of NB.1.8.1 incorporates genetic segments from both XBB and JN.1 [58], two major previously circulating variants. This intricate recombinant architecture has likely endowed NB.1.8.1 with a unique set of mutations that may enhance its transmissibility and immune evasion properties. Such genomic reassortment, facilitated by co-infection and genetic exchange within host cells, represents a key mechanism driving the continuous evolution of SARS-CoV-2, and NB.1.8.1 exemplifies the outcome of this process.

Although NB.1.8.1’s most recent phylogenetic ancestor is XDV.1.5.1, broader phylogenetic analysis indicates that it shares an earlier common ancestor with the currently dominant JN.1 lineage, which itself evolved from BA.2.86 and achieved global prevalence due to heightened transmissibility and immune escape [59]. The relationship between NB.1.8.1 and JN.1 is not one of direct descent but rather that of collateral lineages or evolutionary “cousins” (see Figure 3). Supporting this, comparative genomic studies reveal that NB.1.8.1 lacks several signature non-spike mutations characteristic of BA.2.86 and its direct descendants—such as N:Q229K and ORF1a:A211D—suggesting an independent evolutionary trajectory involving parallel evolution, convergent mutation, or multiple recombination events from common ancestral nodes [60]. In light of its increasing global detection frequency and the presence of mutations associated with altered viral behavior, the World Health Organization (WHO) officially designated NB.1.8.1 as a Variant Under Monitoring (VUM) on 23 May 2025. A VUM is defined as a variant possessing genetic changes suspected to influence viral characteristics such as transmissibility, pathogenicity, or immune evasion but whose epidemiological impact remains uncertain and necessitates enhanced surveillance and evaluation. This classification underscores the imperative for global public health agencies to intensify genomic monitoring of NB.1.8.1, track its spatiotemporal dissemination, and evaluate its potential effects on existing diagnostics, therapeutic agents, and vaccine efficacy.

## 5. Biological Characteristics and Functional Implications

In the mutation profile of the NB.1.8.1 spike protein, Q493E (glutamine to glutamic acid substitution) represents a key mutation of considerable interest [61]. Located in the receptor-binding domain (RBD), this mutation is implicated in modulating immune evasion and infectivity. Studies suggest that the Q493E substitution may alter the local conformation and electrostatic surface of the RBD, thereby reducing antibody binding and facilitating escape from humoral immunity. Furthermore, this mutation may enhance the affinity of the spike protein for ACE2 and consequently increase viral infectivity by optimizing electrostatic interactions through the introduction of a negatively charged residue and by inducing subtle conformational adjustments in the receptor-binding domain that improve steric complementarity. Notably, Q493E is not unique to NB.1.8.1; its independent emergence in multiple JN.1 sublineages indicates a possible convergent adaptive advantage in the current evolutionary landscape [62,63]. Within NB.1.8.1, it is likely that Q493E acts synergistically with other mutations to enhance viral transmission [59,60].

A435S (alanine to serine substitution), another significant mutation in the RBD of NB.1.8.1, exerts a dual effect on immune evasion and receptor binding [64,65]. Functional studies indicate that A435S can significantly reduce the neutralizing efficacy of antibodies targeting conserved epitopes. Concurrently, it may strengthen interactions with ACE2, potentially increasing the efficiency of host cell entry. In NB.1.8.1, A435S, together with Q493E and V445H, forms a functional cluster of mutations that collectively enhance immune evasion and transmission fitness, likely contributing to the variant’s global dissemination.

Beyond these, NB.1.8.1 carries additional spike mutations including T22N, F59S, G184S, F456L, and T478I, totaling seven amino acid changes relative to JN.1. Among these, T22N and F59S are situated in the N-terminal domain (NTD) and may contribute to evasion of NTD-directed neutralizing antibodies. G184S, F456L, and T478I are located within the RBD; the T478I mutation, in particular, has been associated with escape from Class I/II antibodies. The cumulative effect of these mutations considerably alters the spike protein’s structure and antigenicity, conferring a distinct biological advantage. Of particular note is the F456L mutation (phenylalanine to leucine substitution), which belongs to the so-called “FLip” mutation family [66]. Typically, F456L co-occurs with L455F to markedly enhance RBD-ACE2 binding affinity, offsetting potential fitness costs associated with immune-evasive mutations. Although F456L appears alone in NB.1.8.1, it may still positively influence receptor binding. Its presence also raises the possibility of future acquisition of L455F, which could further augment viral transmission, warranting continued monitoring of its evolutionary trajectory. Overall, the mutation profile of the NB.1.8.1 spike protein reflects a functional balance between enhanced binding affinity to the human ACE2 receptor and significantly increased immune evasion capabilities. Multiple studies have demonstrated that NB.1.8.1 exhibits stronger ACE2 binding affinity compared to XFG and most other variants [36,67]. This high affinity enables more efficient infection of host cells, underlying its high transmission potential. Concurrently, mutations such as A435S, Q493E, and T478I alter antigenic epitopes within the receptor-binding domain (RBD), reducing the efficacy of various neutralizing antibodies and facilitating increased spread among populations with existing immunity. The combination of high infectivity and pronounced immune evasion is key to the rapid global emergence and competitive advantage of NB.1.8.1 over other variants. These dual characteristics allow it to overcome host immune barriers, promoting effective infection and transmission.

The numerous spike protein mutations carried by NB.1.8.1, particularly those located in the RBD, substantially impact its recognition by neutralizing antibodies. A study using pseudoviruses and serum from individuals with BA.5 breakthrough infection (infected with JN.1 or XDV+F456L) showed that the neutralization efficiency against NB.1.8.1 was reduced by 1.5- to 1.6-fold compared to LP.8.1.1 [68]. This suggests that despite the presence of antibodies targeting variants such as JN.1, their neutralizing potency against NB.1.8.1 is diminished. Another study also indicated that NB.1.8.1 has gained functional advantages through the accumulation of antibody escape mutations [66]. These findings imply that the emergence of NB.1.8.1 may challenge the protective efficacy of vaccines designed based on earlier variants like JN.1. However, current studies also show that booster immunization with monovalent KP.2 or LP.8.1 mRNA vaccines can still elicit neutralizing antibodies against NB.1.8.1 in mice [36], although with slightly lower titers compared to the homologous antigen. This supports the notion that updating vaccine antigen components remains a viable strategy to counter emerging variants.

Compared to its progenitor-XDV.1, NB.1.8.1 exhibits further enhanced immune evasion capabilities. This is primarily attributed to the acquisition of additional mutations (see Figure 4) such as A435S [63,64], which are strongly associated with immune escape, on the basis of the XDV.1 variant. The A435S mutation has been demonstrated to impair the efficacy of certain class I and class I/IV neutralizing antibodies. The Q493E mutation is also believed to contribute to immune evasion [69,70]. The cumulative effect of these mutations leads to more significant alterations in the antigenic surface of the receptor-binding domain (RBD) of NB.1.8.1, enabling it to more effectively evade antibody responses induced by vaccination or prior infection. This strategy of accumulating multiple immune escape mutations to enhance overall evasion capability is a common pattern in the ongoing evolution of SARS-CoV-2 as it adapts to population immune pressures.

While the spike protein serves as the primary target for neutralizing antibodies and is a critical determinant of viral cell entry, mutations in non-structural proteins (NSPs), structural proteins (nucleocapsid, membrane, and envelope), and accessory proteins also play important roles in viral fitness, replication efficiency, innate immune evasion, and pathogenicity. The NB.1.8.1 genome, derived from its recombinant ancestors XDE and JN.1, harbors a suite of mutations outside the spike gene that may collectively modulate its overall phenotypic characteristics. As a key complex responsible for viral RNA synthesis, the replication–transcription complex (RTC) represents a major site for adaptive mutations [71]. In NB.1.8.1, the RNA-dependent RNA polymerase (NSP12) retains the commonly observed P323L substitution, which has been associated in earlier variants with increased mutation rates and enhanced replication fidelity. Furthermore, the proofreading enzyme NSP14 carries the I42V mutation within its exonuclease domain [72]. This variant, inherited from the JN.1 lineage and highly conserved, may fine-tune the viral mutation rate to balance evolutionary adaptability and genomic stability. Concurrently, variations in NSP13 may influence its ATP hydrolysis and unwinding activities, potentially affecting the overall efficiency of the RTC [73]. In terms of innate immune antagonism, mutations in NB.1.8.1 contribute to enhanced immune evasion. The papain-like protease (PLpro) of NSP3 [74,75], which cleaves viral polyproteins and removes ISG15 modifications from host proteins to suppress interferon responses, carries the S1183L mutation in NB.1.8.1. This mutation may strengthen its inhibitory effect on the type I interferon signaling pathway, thereby improving viral replication in interferon-competent cells. Additionally, ORF6—a key suppressor of interferon signaling that acts by blocking nucleocytoplasmic transport—contains the characteristic D61L mutation present in JN.1 and its sublineages including NB.1.8.1 [76,77]. This substitution may further enhance the ability of ORF6 to antagonize interferon-stimulated gene expression, providing a significant adaptive advantage during early infection. Regarding structural proteins, the nucleocapsid protein is involved not only in RNA packaging but also in the regulation of various host cellular processes. The frequently observed R203K and G204R mutations have been shown to enhance subgenomic RNA transcription and virion assembly efficiency [78], potentially leading to increased viral loads and transmissibility. Moreover, NB.1.8.1 carries the distinctive Q229K mutation within its RNA-binding domain, suggesting a possible role in modulating RNA-binding affinity or protein oligomerization, which could influence virion assembly efficiency and structural stability. Among accessory proteins, ORF3a and ORF8 are primarily involved in regulating host immune responses and apoptosis. ORF3a can induce apoptosis and activate the inflammasome [79], while ORF8 contributes to the downregulation of major histocompatibility complex class I (MHC-I) expression. Functional variations in these proteins may collectively influence viral pathogenicity, inflammatory responses, T cell recognition, and immune evasion potential. In summary, NB.1.8.1 has evolved from the genetic background of the JN.1 lineage, which already possessed a highly adapted non-spike genome [80,81]. A key phylogenetic distinction is that NB.1.8.1 lacks several signature non-spike mutations found in other JN.1 sublineages, such as ORF1a:A211D, indicating a unique evolutionary trajectory. Although the functional impact of most non-spike mutations is relatively subtle, they exhibit clear synergistic effects: mutations in the RTC may collectively optimize viral replication efficiency in the human respiratory tract, while those in immune antagonist proteins such as ORF6 and NSP3 may weaken host innate immune responses, thereby creating a more favorable intracellular environment for the virus. This collective adaptation across the non-spike genome acts in concert with the immune evasion advantages conferred by spike protein mutations, ultimately contributing to the transmission fitness and epidemiological success of NB.1.8.1 in immunized populations.

## 6. Clinical and Epidemiological Perspectives

The NB.1.8.1 variant has emerged as a significant SARS-CoV-2 lineage globally, demonstrating a notable growth advantage in its transmission dynamics. Analysis based on genomic surveillance data from the Global Initiative on Sharing All Influenza Data (GISAID) and fitted epidemiological models indicates that its spread is accelerating across multiple regions. Between April and May 2025, its global prevalence in sequenced samples increased from 3.1% to 15.4%, reflecting substantial competitive transmission fitness, particularly in Asia where it has become the dominant strain.

Transmissibility, quantified by key epidemiological metrics such as the basic reproduction number (R0) and effective reproduction number (Re), is crucial for assessing viral spread. Although precise R_0_ estimates for NB.1.8.1 are not yet available, its increasing detection rate worldwide suggests higher transmission efficiency compared to other co-circulating variants. Serological evidence indicates an approximately 1.5–1.6-fold reduction in neutralizing antibody titers [36,82], implying that immune escape may contribute to its transmission advantage. Wastewater surveillance has corroborated this trend; for example, a report from New Zealand in May 2025 noted that NB.1.8.1 accounted for 21.6% of detected variants, underscoring its current vigorous growth trajectory. From a molecular perspective, NB.1.8.1 carries several key mutations distinct from predecessor lineages. These mutations may enhance binding affinity to the human ACE2 receptor and reduce antibody neutralization capacity, facilitating the variant’s rapid rise to dominance across regions. Collectively, these genetic changes likely confer a selective advantage, enabling stronger immune escape and greater transmissibility compared to contemporaneous lineages [83]. Another notable feature of NB.1.8.1’s transmission dynamics is regional heterogeneity. Surveillance data indicate varying viral fitness across different regions in China, with higher viral activity in southern provinces relative to northern ones, potentially due to differences in climate, demographics, and immune backgrounds [84]. It should be noted that the stochastic nature of sequence submission may introduce bias into the analysis of north–south distributional differences. Similar geographical variation has been observed in Australia, with significant disparities in infection rates between South Australia and Victoria. Consequently, the dissemination of the virus is shaped by a combination of biological determinants, regional contexts, and public health measures. As quantified by the key risk assessment metrics for the NB.1.8.1 variant in Table 1, these factors collectively determine its transmission dynamics.

Regarding pathogenicity—another key indicator in variant epidemiology—most authoritative institutions indicate that although NB.1.8.1 exhibits significantly enhanced transmissibility and immune evasion compared to other circulating variants, it does not lead to a substantial increase in disease severity [88]. Overall, clinical manifestations associated with NB.1.8.1 are relatively mild and resemble those of prior JN.1 infections. Data from the Chinese Center for Disease Control and Prevention (CDC) show that despite its dominance in many countries and regions, the number of fever clinic visits and severe cases remains below historical peaks, without significant fluctuations. Healthcare systems have remained stable, with clinical presentations primarily being mild or asymptomatic. A similar pattern has been observed in Taiwan, China, with no marked increase in intensive care unit admissions or mortality. Although some regions reported short-term rises in infections and hospitalizations, this is likely attributable to higher infection incidence rather than increased intrinsic virulence. Common symptoms include sore throat, fatigue, fever, cough, myalgia, and nasal congestion, occasionally accompanied by gastrointestinal symptoms—consistent with previous Omicron infections [97,98,99]. It is important to note, however, that high-risk groups such as the elderly, individuals with chronic diseases, and immunocompromised patients may still experience severe outcomes. Currently, there is no direct evidence indicating that NB.1.8.1 causes more severe disease. Furthermore, the variant’s susceptibility to existing antiviral drugs remains largely unchanged [82]; no significant alterations associated with drug resistance have been identified in the NB.1.8.1 genome. In other words, current antiviral therapies are expected to retain some protective efficacy. It is important to acknowledge that gaps remain in global data sharing and surveillance. The World Health Organization (WHO) emphasizes that current assessments of NB.1.8.1’s transmissibility, immune escape, and severity are based on limited evidence, highlighting the need for enhanced international collaboration and systematic monitoring. From a population immunity perspective, NB.1.8.1 demonstrates partial immune escape, as evidenced by reduced neutralization titers in convalescent and vaccinated individuals. However, antigenic cartography classifies it within the same antigenic cluster as JN.1, suggesting that vaccines based on JN.1 or LP.8.1 may retain cross-protective potential [100].

As an important descendant lineage of Omicron, NB.1.8.1 exhibits evolutionary trends in clinical manifestations and epidemiological characteristics that align with those of currently circulating variants. Specifically, this variant shows further increases in transmission speed and immune evasion capability, while its intrinsic pathogenicity does not appear to be enhanced—on the contrary, multiple clinical indicators suggest reduced virulence. This phenomenon is consistent with the pattern of convergent evolution observed in recent SARS-CoV-2 evolution, wherein independent branches acquire similar mutations that enhance viral transmissibility in the human population while attenuating pathogenicity. From an adaptive standpoint, enhancing infectivity while reducing host pathogenicity helps expand transmission range and promote long-term host coexistence—an evolutionary strategy observed in many respiratory viruses. Notably, improvements in viral transmission efficiency are often closely associated with mutations in the spike protein, particularly amino acid substitutions in the receptor-binding domain. These changes may enhance the virus’s binding affinity to the human ACE2 receptor or increase replication efficiency. In terms of immune evasion, accumulated mutations in NB.1.8.1 may enable partial escape from neutralizing antibody responses established through prior infection or vaccination, thereby increasing the risk of reinfection—particularly among older adults and immunocompromised groups, where breakthrough infection rates may rise [101,102].

The progressive discontinuation of systematic SARS-CoV-2 surveillance in many regions represents a significant shift from broad genomic monitoring to a more targeted approach focused only on variants flagged by global health agencies as having potential elevated transmissibility or pathogenicity. While this transition may alleviate governmental financial burdens and reduce public anxiety, it carries substantial implications for global outbreak preparedness and variant tracking. The reduction in routine surveillance compromises the completeness and timeliness of sequence data submitted to international repositories such as GISAID. Such data are critical for detecting emerging variants, evaluating their transmission dynamics, and assessing immune evasion potential. In the absence of systematic testing and sequencing, variants with moderate but biologically significant fitness advantages may circulate undetected until localized outbreaks occur. A notable example is the XDV.1 lineage, which did not cause widespread transmission globally but led to significant regional waves in China—highlighting how heterogeneous surveillance can obscure the true spread and impact of a variant. Furthermore, disparities in population immunity—driven by differing vaccination histories, infection backgrounds, and booster uptake—create conditions in which new variants can exploit immunological gaps. Delayed detection increases the risk of resurgent waves, particularly among immunologically vulnerable subgroups.

Although current pathogenicity data for NB.1.8.1 indicate reduced clinical severity compared to earlier strains—including lower rates of severe disease and hospitalization—the possibility of highly pathogenic recombinant strains emerging in subsequent evolutionary stages cannot be ruled out. Viral evolution is highly unpredictable, and factors such as genetic recombination, accumulation of point mutations, and zoonotic transmission events may contribute to the emergence of novel phenotypes. Should a variant simultaneously acquire enhanced transmissibility, significant immune escape, and increased virulence, it could potentially trigger a new wave of infections accompanied by more severe clinical manifestations, thereby placing substantial strain on public healthcare systems. Therefore, systematic molecular epidemiological research on emerging variants such as NB.1.8.1 is critically important. This includes continuous monitoring of viral genomes, dynamic tracking of mutation frequencies, and analysis of regional distribution patterns. In addition, assessing immune escape potential using reverse genetics techniques, pseudovirus neutralization assays, and other methods is essential for evaluating vaccine efficacy and the level of immunity conferred by prior infection. Risk assessment efforts should prioritize dynamic and individualized evaluations of high-risk populations, including the elderly, individuals with chronic underlying conditions, and immunocompromised patients. These groups are not only more susceptible to severe outcomes following infection but may also face higher reinfection risks due to inadequate immune memory.

From a long-term epidemiological perspective, refining public health strategies in response to SARS-CoV-2 variants requires multidisciplinary collaboration and global data sharing. Key measures include promoting the development of broad-spectrum vaccines, strengthening the stockpiling and application of antiviral drugs, optimizing tiered diagnosis and treatment systems, and enhancing societal emergency response capacities in the face of evolving outbreaks. Only through comprehensive prevention and control measures, informed by robust scientific analysis, can public health be effectively protected and the overall societal impact of viral evolution be mitigated.

## 7. Discussion

This review synthesizes current evidence on the SARS-CoV-2 variant NB.1.8.1, outlining well-established characteristics as well as persistent knowledge gaps. It is established that NB.1.8.1 exhibits a competitive advantage in transmission over other co-circulating lineages, likely driven by a combination of enhanced intrinsic transmissibility and immune escape—supported by serological studies indicating reduced neutralization antibody titers. Preliminary molecular profiling has identified key mutations that may contribute to these phenotypic features. Importantly, available data from multiple regions consistently indicate that its increased transmissibility has not been associated with a significant rise in disease severity or notable changes in clinical manifestation compared to prior Omicron sublineages. Nevertheless, considerable uncertainties remain. There is currently a lack of precise quantification of its basic reproduction number (R_0_), which hinders direct and accurate comparative modeling of its transmission dynamics. The specific contributions and synergistic mechanisms of its mutations to infectivity and immune evasion have not yet been fully elucidated through functional assays. Furthermore, the long-term epidemiological trajectory of NB.1.8.1 within diverse population-level immunity backgrounds—particularly in regions with varying vaccination and infection histories—requires further evaluation. A major strength of the current evidence lies in robust global genomic surveillance systems, such as GISAID, which enable near real-time tracking of variant prevalence. Ancillary data from wastewater surveillance and serosurveys are also accumulating, providing valuable insights into transmission dynamics even in the context of reduced clinical testing. This review integrates these multi-source data to present a preliminary consolidated assessment of NB.1.8.1.

However, several limitations must be acknowledged. The available data are subject to biases inherent in uneven sequencing coverage, fluctuating testing policies, and shifts in surveillance strategies. Evidence regarding clinical severity is largely derived from syndromic surveillance and routine health metrics; the absence of controlled cohort studies adjusting for confounders such prior immunity impedes definitive conclusions on intrinsic virulence. Although comprehensive, this review is also constrained by the preliminary nature of much of the source material, which includes preprints and rapid communications, thereby limiting the certainty of the conclusions.

The emergence and evolution of NB.1.8.1 underscore the indispensability of sustained and collaborative genomic and epidemiological surveillance systems. Its classification as a Variant Under Monitoring (VUM) reflects a prudent risk-aware strategy. Continuous monitoring is essential for the early detection of any significant changes in viral antigenicity, transmission dynamics, or clinical severity. Such systems enable proactive public health responses, informing potential updates to vaccine formulations, guiding the use of existing antiviral therapies, and helping healthcare systems prepare for potential case surges. Maintaining vigilance against variants like NB.1.8.1 is fundamental to the long-term management of COVID-19, ensuring that containment strategies evolve in tandem with viral changes.

## Figures and Tables

**Figure 1 viruses-17-01457-f001:**
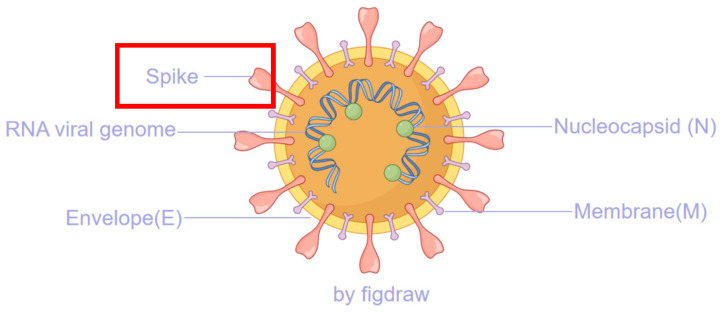
The major structural proteins of SARS-CoV-2, with the spike protein highlighted in a red box, play a critical role in the viral replication process.

**Figure 2 viruses-17-01457-f002:**
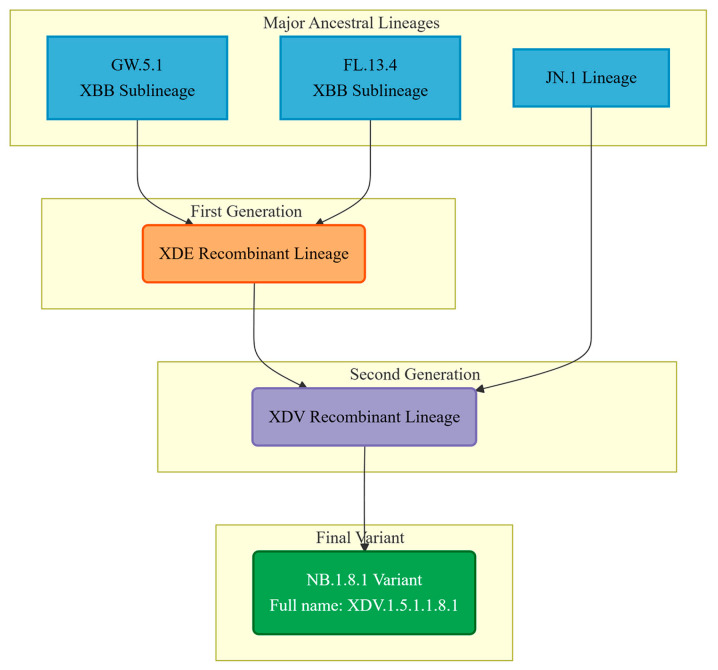
The flowchart illustrates the origins of the different genomic regions of the NB.1.8.1 variant and its formation process as a recombinant.

**Figure 3 viruses-17-01457-f003:**
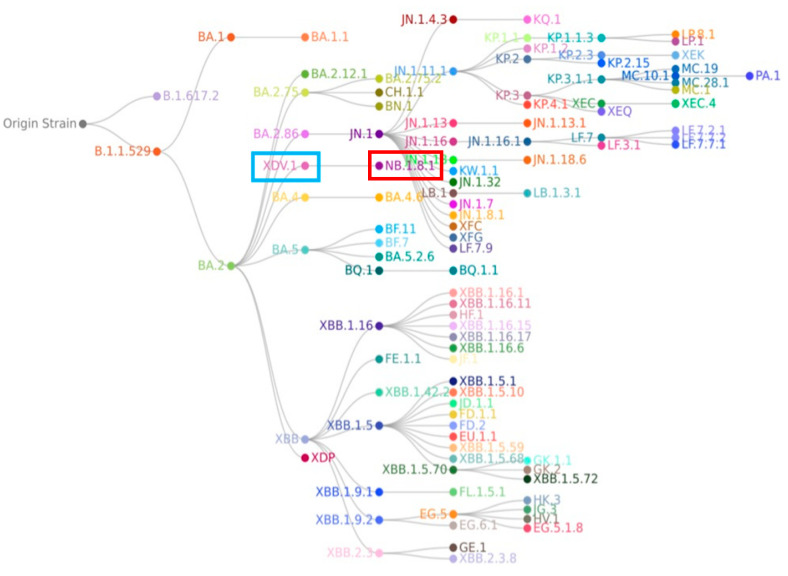
In the phylogenetic tree of the SARS-CoV-2 NB.1.8.1 lineage, the XDV.1 and NB.1.8.1 variants are highlighted in blue and red, respectively. Phylogenetic analysis reveals that NB.1.8.1 is a descendant of the Omicron lineage, but it forms a distinct branch separate from the JN.1 variant and is not a direct descendant of JN.1.

**Figure 4 viruses-17-01457-f004:**
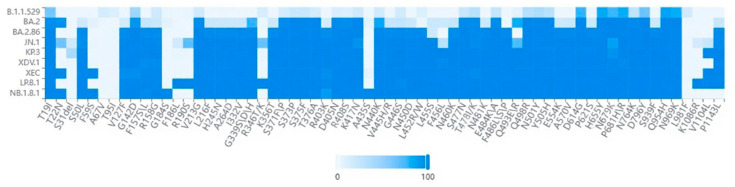
Heatmap analysis across various lineages. Key amino acid mutations in the NB.1.8.1 lineage identified through spike protein sequence analysis of GISAID submissions. The dataset comprised 500 high-quality genomic sequences for each variant. These sequences were selected from multiple databases, including GISAID and GenBank, based on stringent quality criteria, completeness, and their collection date between December 2024 and September 2025. Furthermore, the selection encompassed sequences sourced from diverse countries and regions.

**Table 1 viruses-17-01457-t001:** Risk-assessment profile of the SARS-CoV-2 NB.1.8.1 variant.

Risk Assessment Indicators	Assessment Results
Disease severity	No substantial change; similar to previously circulating Omicron sub-variants [83,85]
ICU admission/hospitalization rate	No significant increase; remained below all preceding peaks [86]
Predominant clinical manifestation	Sore throat, fatigue, fever, mild cough, myalgia, and nasal congestion [87,88]
Vaccine effectiveness	Current vaccines retain high effectiveness against severe disease [67,68]
Antiviral drug efficacy	No well-defined resistance-associated mutations detected [89,90]
Transmissibility	Moderate to high [36,66]
Immune escape	Moderate [67,91]
Health-system impact	Low to moderate [86]
Effectiveness of countermeasures	High [92,93,94,95,96]

## Data Availability

Not applicable.

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
