# Peer review of "Global Surveillance and Biological Characterization of the SARS-CoV-2 NB.1.8.1 Variant: An Emerging VUM Lineage Under Scrutiny"

_viruses, 2025, doi:10.3390/v17111457_

Round 1

Reviewer 1 Report

Comments and Suggestions for Authors

This reviewer likes the idea of providing an overview of the phenotypic and genetic characteristics, morbidity, mortality, and transmission of SARS-CoV-2 variants currently in circulation.

However, this reviewer has major concerns :

  • The title suggests that this is a comprehensive review of SARS-CoV-2 NB.1.8.1, but the review actually deals exclusively with changes in the spike gene. This is certainly important, but it should then be clearly evident from the title. Otherwise, changes in other genes (after all, the remaining almost 90% of the viral genome) should also be addressed.
  • Furthermore, the introduction seems incomplete or could be misleading: the paragraph on the error-prone nature of viral RNA-dependent RNA polymerases: coronaviruses do indeed have proofreading enzymes (e.g., nsp14), thus the authors should check the literature here.
  • Further references should be cited to support statements (e.g., lines 103-108, where references appear to be missing).
  • The review should clearly address the fact that systematic surveillance has been discontinued in many regions. Which impact could that have? (GSAID requires data). What do the authors propose?
  • Table 1 is missing a caption and references. What are the assessment results cited here based on? Compared to what? 

Author Response

Comments 1: The title suggests that this is a comprehensive review of SARS-CoV-2 NB.1.8.1, but the review actually deals exclusively with changes in the spike gene. This is certainly important, but it should then be clearly evident from the title. Otherwise, changes in other genes (after all, the remaining almost 90% of the viral genome) should also be addressed.

Response 1: This paper is a comprehensive review of the SARS-CoV-2 NB.1.8.1 variant. The initial draft overemphasized the description of the spike protein, while the discussions of other structural, non-structural, and accessory proteins were relatively neglected. To address this limitation, the revised manuscript has systematically incorporated the relevant content, which has been highlighted in red for your convenience. This revision is a direct response to the first reviewer's comment.

Comments 2: Furthermore, the introduction seems incomplete or could be misleading: the paragraph on the error-prone nature of viral RNA-dependent RNA polymerases: coronaviruses do indeed have proofreading enzymes (e.g., nsp14), thus the authors should check the literature here.

Response 2: In the Introduction, we noted that the RNA-dependent RNA polymerase (RdRp) utilized by most RNA viruses is generally highly error-prone. However, SARS-CoV-2 (COVID-19) possesses unique replication proofreading activity, conferred by specific proteins such as non-structural protein 14 (NSP14) within its replication complex. In the revised manuscript, we have elaborated on this specific mechanism based on a review of relevant literature, with all newly added content highlighted in red.

Comments 3:Further references should be cited to support statements (e.g., lines 103-108, where references appear to be missing).

Response 3: References have been added to content in Lines 103-108 of the manuscript.

Comments 4:The review should clearly address the fact that systematic surveillance has been discontinued in many regions. Which impact could that have? (GSAID requires data). What do the authors propose?

Response 4: As routine SARS-CoV-2 testing has now been suspended worldwide, only disease control centers and some databases continue to monitor potential variants identified by the WHO. Consequently, the impacts of this shift are complex and twofold. We have systematically addressed this in the revised manuscript by elucidating the associated benefits and drawbacks, and have clearly marked the proposed recommendations.

Comments 5:Table 1 is missing a caption and references. What are the assessment results cited here based on? Compared to what? 

Response 5: References have been added to Table 1. The evaluation results are primarily based on a comparative analysis of the characteristics of the Omicron variant and some earlier strains. Furthermore, some of the assessed outcomes are derived from a dedicated WHO report on the NB.1.8.1 lineage, which ensures the scientific rigor of the findings.

Reviewer 2 Report

Comments and Suggestions for Authors

The submitted manuscript is a timely and comprehensive review dedicated to characterizing the novel SARS-CoV-2 variant NB.1.8.1, classified by the World Health Organization (WHO) as a Variant Under Monitoring (VUM). The topic's relevance is undeniable, given the continuous evolution of the virus and the persistent challenges for global public health. The work fulfills an important function by systematizing and synthesizing disparate data from genomic surveillance, laboratory studies, and epidemiological reports, offering a consolidated assessment of the potential risks associated with NB.1.8.1.

Strengths of the Manuscript

The authors consistently guide the reader from the general evolutionary context of SARS-CoV-2 to the specifics of the NB.1.8.1 variant. The discussion of its origin in the context of recombination (XDV and JN.1 lineages) is particularly valuable, as it reflects the current understanding of the driving forces behind viral evolution.

The section on biological characteristics provides a detailed and competent breakdown of the functional consequences of key spike protein mutations (A435S, Q493E, F456L, T478I, etc.). The explanation of their role in enhancing ACE2 affinity and immune evasion is based on up-to-date literature.

The manuscript successfully balances data from genomic surveillance (GISAID), results from in vitro studies (pseudoviruses, neutralization), and epidemiological observations. The integration of these diverse sources significantly enhances the review's value.

The provided risk assessment profile table is an extremely useful tool for visualizing and quickly evaluating the threat posed by NB.1.8.1. The conclusions about increased transmissibility in the absence of increased disease severity are consistent with current observational data.

However, there are some comments:

The article contains numerous punctuation and stylistic errors and requires careful and thorough editing.

May be accepted for publication after extensive editing and re-review.

Author Response

Comments 1: The article contains numerous punctuation and stylistic errors and requires careful and thorough editing.

Response 1: The article has already been revised.

Reviewer 3 Report

Comments and Suggestions for Authors

The review focuses on the NB.1.8.1 lineage while providing a clear and informative introduction that outlines the history of the major SARS-CoV-2 variants of concern (VOCs), their characteristic properties, and key genomic sites where mutations have arisen convergently across multiple viral lineages. These substitutions are commonly linked to enhanced ACE2 receptor affinity, increased transmissibility, and evasion of neutralizing antibodies. The authors also emphasize the pivotal role of recombination in SARS-CoV-2 evolution. The emergence of the XDV variant, similar to the previously highly successful recombinant lineage XBB, illustrates this process. Overall, systematizing the available information on the NB.1.8.1 SARS-CoV-2 variant in a review format is both timely and valuable.

Comment 1. Line 117. “increased effective reproduction number (Re)” Please provide the reference

Comment 2. Line 213-214 “This observation is supported by animal studies showing lower viral loads in the lungs and reduced tissue damage compared to XBB sublineages” Please provide the reference

Comment 3. Lines 174-181 and lines 219-220. The role of the L455S substitution is discussed twice in the manuscript. Please consider whether both mentions are necessary, or if this information could be consolidated to avoid redundancy.

Comment 4. Lines 228-230. “First detected in late 2024 through genomic surveillance systems, NB.1.8.1, a descendant of the recombinant XDV lineage, attracted attention due to its rapid expansion in certain regions despite low global prevalence”. Please specify which regions are referred to here.

Comment 5. Line 250 “The GISAID database was specifically queried”. Please specify the query parameters used for the GISAID search. Due to incomplete or partial sequences, some viruses may not be correctly identified by automated lineage recognition systems. Clarifying the search criteria would enhance transparency and reproducibility.

Comment 6. Line 272. Consider adding a schematic figure illustrating the origin of different genomic regions of the NB.1.8.1 variant. Such a visual representation would greatly facilitate readers’ understanding of the recombinant structure and evolutionary context.

Comment 7. Line 314-315. “it is likely that Q493E acts synergistically with other mutations to enhance viral transmission”. Please provide the reference for this statement.

Comment 8. Lines 340-341 “Multiple studies have demonstrated that NB.1.8.1 exhibits stronger ACE2 binding affinity compared to XFG and most other variants”. Please provide the references.

Comment 9. Lines 356-357. “Another study also indicated that NB.1.8.1 has gained functional advantages through the accumulation of antibody escape mutations”. Please provide the reference.

Comment 10. Line 378. Figure 3. If this figure is based on analyses performed by the authors, please indicate the number of genomes included in the dataset and specify the quality criteria applied for sequence selection. Providing these details would improve the methodological transparency of the study.

Comment 11. Line 390. “effective reproduction number (Rt)” in Line 117 Re was used for effective reproduction number. Please ensure the consistent use of the terms and designations.

Comment 12. Line 391. The manuscript states that NB.1.8.1 is characterized by an “increased effective reproduction number (Re)” (line 117), while later noting that “precise R₀ estimates for NB.1.8.1 are not yet available” (and again in lines 505–506, “There is currently a lack of precise quantification of its basic reproduction number [R₀]”). Please clarify this apparent inconsistency and specify whether the authors refer to R₀, Re, or both, and how these estimates were derived or inferred.

Comment 13. Lines 404-410. Please provide the references or methodological details supporting the conclusions regarding geographical heterogeneity. It would also be important to discuss whether uneven sampling or undersampling in certain regions could have influenced these observations.

Comment 14. Line 410. Consider adding transition phrases to improve the flow within the section discussing the risk assessment. Additionally, it may be clearer to introduce Table 1 directly in the text rather than in parentheses, to better integrate it into the narrative.

Comment 15. Line 433-434. “Furthermore, the variant’s susceptibility to existing antiviral drugs remains largely unchanged”. Please provide the references

Author Response

Comments 1: Line 117. “increased effective reproduction number (Re)” Please provide the reference

Response 1: References have been added.

Comments 2:Line 213-214 “This observation is supported by animal studies showing lower viral loads in the lungs and reduced tissue damage compared to XBB sublineages” Please provide the reference

Response 2: References have been added.

Comments 3:Lines 174-181 and lines 219-220. The role of the L455S substitution is discussed twice in the manuscript. Please consider whether both mentions are necessary, or if this information could be consolidated to avoid redundancy.

Response 3: Upon careful review of the manuscript, we identified a duplicated passage. After thorough consideration, we have removed one of the instances to eliminate redundancy. This revision has been carefully executed to ensure the integrity and flow of the original text are fully preserved.

Comments 4: Lines 228-230. “First detected in late 2024 through genomic surveillance systems, NB.1.8.1, a descendant of the recombinant XDV lineage, attracted attention due to its rapid expansion in certain regions despite low global prevalence”. Please specify which regions are referred to here.

Response 4: The manuscript has been supplemented in specific sections for clarity and completeness. Please refer to the red-highlighted areas for the detailed revisions.

Comments 5:Line 250 “The GISAID database was specifically queried”. Please specify the query parameters used for the GISAID search. Due to incomplete or partial sequences, some viruses may not be correctly identified by automated lineage recognition systems. Clarifying the search criteria would enhance transparency and reproducibility.

Response 5: The search parameters used for the GISAID database have been incorporated into the revised manuscript.

Comments 6:Line 272. Consider adding a schematic figure illustrating the origin of different genomic regions of the NB.1.8.1 variant. Such a visual representation would greatly facilitate readers’ understanding of the recombinant structure and evolutionary context.

Response 6:We agree with the reviewer's comment on the need for a flowchart to delineate the recombinant origin of the NB.1.8.1 lineage. Accordingly, this visual aid has been created and included in the revised manuscript.

Comments 7:Line 314-315. “it is likely that Q493E acts synergistically with other mutations to enhance viral transmission”. Please provide the reference for this statement.

Response 7:References have been added.

Comments 8: Lines 340-341 “Multiple studies have demonstrated that NB.1.8.1 exhibits stronger ACE2 binding affinity compared to XFG and most other variants”. Please provide the references

Response 8:References have been added.

Comments 9:Lines 356-357. “Another study also indicated that NB.1.8.1 has gained functional advantages through the accumulation of antibody escape mutations”. Please provide the reference.

Response 9:References have been added.

Comments 10: Line 378. Figure 3. If this figure is based on analyses performed by the authors, please indicate the number of genomes included in the dataset and specify the quality criteria applied for sequence selection. Providing these details would improve the methodological transparency of the study.

Response 10:This figure is based on our original analysis. The relevant details of the dataset used have been added to the revised manuscript.

Comments 11: Line 390. “effective reproduction number (Rt)” in Line 117 Re was used for effective reproduction number. Please ensure the consistent use of the terms and designations.

Response 11:The identified inconsistencies have been addressed in the revised version.

Comments 12: Line 391. The manuscript states that NB.1.8.1 is characterized by an “increased effective reproduction number (Re)” (line 117), while later noting that “precise R₀ estimates for NB.1.8.1 are not yet available” (and again in lines 505–506, “There is currently a lack of precise quantification of its basic reproduction number [R₀]”). Please clarify this apparent inconsistency and specify whether the authors refer to R₀, Re, or both, and how these estimates were derived or inferred.

Response 12:A reviewer raised a question concerning the basic and effective reproduction numbers. It is noted in the literature that the effective reproduction number (Re) of the NB.1.8.1 lineage shows an increasing trend; however, its basic reproduction number (R0) is not explicitly documented. Consequently, it is stated in the manuscript that the R0 has not been directly and precisely quantified. An additional point to note is that the virus source employed in the relevant references was a pseudovirus, not the live virus. While pseudovirus technology is commonly used in studies of viral functional characteristics, findings derived from it are for reference only and should not be equated with the actual properties of the live virus. Thus, in the corresponding description, deliberately cautious wording was adopted, explicitly stating the lack of precise quantitative data for the basic reproduction number. This is my clarification and response to the comment.

Comments 13: Lines 404-410. Please provide the references or methodological details supporting the conclusions regarding geographical heterogeneity. It would also be important to discuss whether uneven sampling or undersampling in certain regions could have influenced these observations.

Response 13:References have been added.

Comments 14:Line 410. Consider adding transition phrases to improve the flow within the section discussing the risk assessment. Additionally, it may be clearer to introduce Table 1 directly in the text rather than in parentheses, to better integrate it into the narrative.

Response 14:Seamless integration of Table 1 into the main body of the text has been achieved through the addition of connecting sentences, replacing its previous presentation as an explicit callout.

Comments 15:Line 433-434. “Furthermore, the variant’s susceptibility to existing antiviral drugs remains largely unchanged”. Please provide the references

Response 15:References have been added.

Reviewer 4 Report

Comments and Suggestions for Authors

This review provides a comprehensive overview of the emerging SARS-COV-2 NB.1.8.1 variant including its biological characterization. The review is well presented and clear. The material discussed is very interesting even though limited by the preliminary nature of some information. The manuscript should be of interest for a broad readership.

Minor revisions:

Lines 306-307: …, a region critical for virus binding to the host ACE2 receptor and for neutralization by antibodies,.This part can be deleted since this concept has been already introduced in the text earlier in the manuscript and won’t  change the sense of the sentence.

How the Q493E enhances binding affinity to ACE2 receptor? Lines 310-311

Insert a reference for the statement included in the lines 367-369

Lines 393-394: insert reference for the serological evidence of reduced neutralizing antibodies titers

Author Response

Comments 1: Lines 306-307: …, a region critical for virus binding to the host ACE2 receptor and for neutralization by antibodies,.This part can be deleted since this concept has been already introduced in the text earlier in the manuscript and won’t  change the sense of the sentence.

Response 1:The relevant section has been condensed for conciseness without altering the core meaning of the original text.

Comments 2:How the Q493E enhances binding affinity to ACE2 receptor? Lines 310-311

Response 2: The mechanism by which Q493E enhances binding affinity to the ACE2 receptor has been added to the revised manuscript and is highlighted in red.

Comments 3:Insert a reference for the statement included in the lines 367-369

Response 3: References have been added.

Comments 4: Lines 393-394: insert reference for the serological evidence of reduced neutralizing antibodies titers

Response 4: References have been added.
